# Relationship between Dietary Total Antioxidant Capacity and the Prognosis of Amyotrophic Lateral Sclerosis

**DOI:** 10.3390/nu14163264

**Published:** 2022-08-10

**Authors:** Jihyun Eom, Bugyeong Son, Seung Hyun Kim, Yongsoon Park

**Affiliations:** 1Department of Food and Nutrition, Hanyang University, 222 Wangsimni-ro, Seongdong-gu, Seoul 04763, Korea; 2Cell Therapy Center, Hanyang University Hospital, 222-1 Wangsimni-ro, Seongdong-gu, Seoul 04763, Korea; 3Department of Neurology, Hanyang University Hospital, 222-1 Wangsimni-ro, Seongdong-gu, Seoul 04763, Korea

**Keywords:** amyotrophic lateral sclerosis, dietary total antioxidant capacity, disease progression rate, event-free survival

## Abstract

Antioxidant intake has been suggested to be associated with the prognosis of amyotrophic lateral sclerosis (ALS). This study aimed to investigate whether dietary total antioxidant capacity (DTAC) and that of major food groups are related to disease progression rate (ΔFS) and survival time in ALS patients. A total of 301 participants diagnosed with sporadic ALS according to the revised El Escorial criteria were recruited from March 2011 and followed up to the event occurrence, or the end of October 2021. Events included percutaneous endoscopic gastrostomy, tracheostomy, and death. DTAC was estimated using task automation and an algorithm based on 24 h dietary recall. ΔFS was negatively correlated with the vegetable and legume DTAC, and event-free survival was different among the tertiles of vegetables and legumes DTAC. Consistently, the risk of events was negatively associated with DTAC from vegetables and legumes. These results suggest that the intake of antioxidants, especially those derived from vegetables and legumes, has a beneficial effect on delaying disease progression and prolonging survival in patients with ALS. Further studies with large prospective cohorts and clinical trials are needed to determine whether the consumption of foods with high DTAC improves the prognosis of ALS.

## 1. Introduction

Amyotrophic lateral sclerosis (ALS) is a fatal progressive neurodegenerative disease characterized by degeneration and loss of motor neurons in the brain and spinal cord, ultimately leading to death from respiratory failure [1]. The management of ALS has evolved in recent years following the efforts of a multidisciplinary team with pharmacological options and nutritional intervention strategies; however, ALS is still an incurable disease [2].

Our previous studies reported that consumption of beta-carotene and fruits rich in antioxidants was inversely related to the risk of developing ALS [3], and that nutritional status was associated with disease progression [4,5,6] and survival of patients with ALS [7]. In addition, the ALS functional rating scale-revised (ALSFRS-R) score was higher in patients with ALS who consumed more vegetables [6]. Nieves et al. [8] also reported that the intake of vegetables and foods high in carotenoids was associated with higher ALSFRS-R scores among patients with ALS. Moreover, treatment with antioxidants, such as flavonoids, vitamin E, and vitamin C, extended the survival rate and increased the muscle function by decreasing the production of reactive oxygen species (ROS) in ALS mice [9]. In cohort studies, regular use of vitamin A and E supplements was inversely associated with shorter survival duration and the risk of death in patients with ALS, but the use of vitamin C supplements was not associated with the prognosis of ALS [10,11]. Furthermore, two previous clinical trials showed that vitamin E supplementation slowed the progression rate according to the ALS Health State scale, but affected neither survival rates [12] nor physical performance, as measured by the Limb Norris Scale [13], in patients with ALS.

The inconsistency in the association between the prognosis of ALS and intake of antioxidants or antioxidant rich foods may be due in part to the synergistic interaction between antioxidants [14]. The dietary total antioxidant capacity (DTAC) represents the sum of antioxidant activity from the diet rather than the effects of individual antioxidants [15]. Previous studies reported that the DTAC was inversely associated with the risk of disease onset, such as stroke [16,17], metabolic syndrome [18], cardiovascular disease [19], and colorectal, gastric, and endometrial cancer [20]. The risk of Parkinson’s disease onset was inversely associated with consumption of beta-carotene and vitamin E, but not with DTAC [21]. However, to our knowledge, there have been few studies to examine the relationship between any disease prognosis and DTAC. Therefore, the present study aimed to investigate whether the total DTAC and that of major food groups are related to disease progression rate and survival time in patients with ALS.

## 2. Methods

### 2.1. Participants

Participants were recruited at the ALS outpatient clinics at Hanyang University Hospital, Seoul, Korea, from March 2011 to October 2020, and followed until the occurrence of events or the end of October 2021, whichever came first. Events included percutaneous endoscopic gastrostomy (PEG), tracheostomy, and death. Participants who were alive for 5 years without these events, or who were lost to follow-up, were censored at the study termination or last visit, respectively.

A total of 522 patients diagnosed as definite, clinically probable, or probable with laboratory-supported sporadic ALS based on the revised EI Escorial criteria (rEEC) were screened for this project (Figure 1). Patients were excluded from the study if they met any of the following criteria: (1) symptom onset occurred more than 2 years previously (*n* = 156); (2) having a PEG (*n* = 45); (3) having a tracheostomy or non-invasive ventilation (NIV) (*n* = 20) at the time of the dietary survey. A total of 301 eligible participants were included in this study.

The study was conducted in accordance with the Declaration of Helsinki, and all procedures were approved by the Institutional Review Board of Hanyang University (HYI-14-105-15). Written informed consent was obtained from all participants prior to enrolment or from caregivers, if the participants were unable to write.

### 2.2. Data Collection

The participant’s medical records were reviewed to obtain information regarding demographic data, diagnosis, age at symptom onset, site of symptom onset, the Korean version of ALSFRS-R scores (K-ALSFRS-R), height, weight, and treatment data. Data regarding PEG insertion, respiratory interventions (such as NIV or tracheostomy), and death were collected during follow-ups. Then, a trained dietitian performed a dietary survey for each participant, which included a 24 h dietary recall, as well as questions concerning exercise, smoking status, drinking habits, and sun exposure.

Dietary information was analyzed by CAN-pro (version 5.0, Korean Nutrition Society, Seoul, Korea). The exercise was defined as at least once a week and drinking at least once a month. The time point of the dietary survey from symptom onset was defined as the time between symptom onset and the dietary survey. The disease progression rate (ΔFS) was calculated using the following formula [22,23]: ΔFS = (48 − ALSFRS-R score at the time of survey)/duration from symptom onset to the time of the survey (months). The treatment data included riluzole and edaravone, which were the only two available ALS treatments approved by the U.S. Food and Drug Administration (FDA).

### 2.3. Estimation of DTAC

The antioxidant capacity of individual antioxidants was expressed as vitamin C equivalents (VCE) using the 2,2′-azino-bis (3-ethylbenzothiazoline-6-sulphonic acid) assay [24]. The TAC database was developed for 42 dietary antioxidants (e.g., vitamins C and E, carotenoids, and flavonoids) based on the Korean Rural Development Administration (KRDA) and U.S. Department of Agriculture (USDA) [25,26,27]. In the case of the same food, the KRDA database was prioritized over the USDA database, and when no matching food was found, values for similar foods were used based on similarities in a botanical group and the composition of nutrients. The database covered 92.15% of food intake.

DTAC was calculated according to the total antioxidant capacity (mg VCE) multiplied by the intake of each food (g) estimated from 24 h recall [28]. DTAC was categorized into the 17 food groups suggested by the Korea National Health and Nutrition Examination Survey: grains, potatoes, sugars and sweets, legumes, nuts and seeds, vegetables, mushrooms, fruits, meats, poultry, eggs, fish and shellfish, seaweeds, milk and dairy products, oils and fats, beverages and alcohols, seasonings, and others [29].

### 2.4. Statistical Analyses

Statistical analyses were performed using SPSS (version 25.0; SPSS Inc., Chicago, IL, USA), and statistical significance was considered at a *p*-value of <0.05. Variables with a normal distribution were analyzed using the Kolmogorov–Smirnov test, and nonparametric variables were analyzed using nonparametric tests. Continuous variables were expressed as means ± standard deviations (SD) and verified using Bonferroni’s post hoc test after one-way analysis of variance (ANOVA) or the Kruskal–Wallis test. Proportions of nominal variables were compared using the chi-square test or Fisher’s exact test.

The correlations between ΔFS and DTAC were calculated using the Spearman’s rank correlation coefficients. The Kaplan–Meier method was used to calculate the cumulative survival probabilities, and the difference between the survival curves was evaluated using the log-rank test. Hazard ratios and 95% confidence intervals were calculated using Cox proportional hazards regression analysis after adjusting for confounding factors. In multivariate models, covariates with a *p*-value < 0.20 were selected as confounding factors (symptom onset age, sex, BMI, site of onset, disease progression, exercise, sun exposure, smoking, drinking, and energy intake). As a result, symptom onset age, sex, BMI, site of onset, disease progression, and energy intake were included in the fully adjusted model [30]. The *p*-value for trend was calculated using the median of tertiles in DTAC, with the lower third serving as the reference group.

## 3. Results

### 3.1. Characteristics of Participants According to the Tertile of DTAC

Participants in the upper third of total DTAC had a higher ALSFRS-R score and longer sun exposure than those in the middle and lower thirds of the total DTAC (Table 1). In addition, participants in the middle and upper thirds of total DTAC exercised more than those in the lower third of the total DTAC. There were no significant differences in the age at symptom onset, sex, bulbar onset, symptom duration, ΔFS, bulbar score, BMI, smoking, drinking, and received treatments (including, riluzole and edaravone) among the tertiles of DTAC of the major food groups (Table 1 and Table 2). The participants within the lower third of the fruit DTAC had a slower ΔFS than those in the middle and upper thirds of the fruit DTAC. However, the participants in the lower third of vegetable and legume DTAC had a more rapid ΔFS than those in the middle and upper thirds of the vegetable and legume DTAC.

### 3.2. DTAC and Disease Progression Rate

The ΔFS was not significantly correlated with total DTAC, but was negatively correlated with vegetable and legume DTAC (Figure 2). The proportions of the major food groups that contributed to DTAC were fruits (43.2%), vegetables (37.2%), and legumes (13.8%). There was no correlation between ΔFS and fruit DTAC.

### 3.3. DTAC and Survival Time

Kaplan–Maier analysis showed that event-free survival was not significantly different among the tertiles of total DTAC during the 5-year follow-up period (Figure 3). However, participants in the middle and upper third of the vegetable and legume DTAC had longer event-free survival than those in the lower third. Consistently, the multivariate-adjusted Cox’s regression analysis showed that the vegetable and legume DTAC, but not the fruit DTAC, were negatively associated with the risk of events (Figure 4).

## 4. Discussion

This study showed that vegetable and legume DTAC was associated with slow progression and event-free survival in ALS patients. However, statistical significance was not observed in the total DTAC and fruit DTAC. In our cohort study, the mean ΔFS was 0.75, the mean ALSFRS-R score was 37, and the mean survival time was 30 months (range, 7–60 months). Similarly, Yu et al. [5] and Baek et al. [31] showed that the mean ΔFS was 0.73–0.75, and the mean ALSFRS-R score was 35–37. The average age of symptom onset was also 50.7–55.9 in the other Korean studies [31,32], which was similar to 54.6 years in the present study. Additionally, the rate of events, including PEG, tracheostomy, and death, was 57.8%, which was similar to the 61.0% result in the previous study [5]. The average DTAC was 358.2 mg VCE/d, which is comparable to the estimated DTAC of 384.7 mg VCE/d from the Korean adults’ diets [33]. Although ALSFRS-R scores differed among the tertiles of DTAC, there were no significant differences in the BMI or bulbar score, which reflected the equal swallowing ability at the time point of the dietary survey.

There have been no previous studies examining the relationship between DTAC and ALS. Previous research on Parkinson’s disease (PD), which is one of the neurodegenerative diseases, showed no significance between the DTAC and the risk of PD [21]. However, the DTAC has been shown to be inversely associated with the risk of stroke onset [16,17], metabolic syndrome [18], cardiovascular disease [19], and colorectal, gastric, and endometrial cancers [20]. In the present study, the DTAC from fruit accounted for 43% of the total DTAC, and our participants consumed 50% of fruits as fruit juice because of eating problems, such as dysphagia. Pase et al. [34] reported that higher fruit juice intake was associated with lower brain volume and poorer memory. A dietary pattern rich in fruits, excluding fruit juice, was positively associated with ALSFRS-R scores in patients with ALS, but fruit juice was not [8]. Our previous study showed that the intake of fruits, including fruit juice, was not significantly associated with the ALSFRS-R score in patients with ALS [6]. Van der Sluis et al. [35] reported that apple juice contained fewer antioxidants, particularly flavonoids, than fresh apples, which were not transferred into the juice and remained in the pulp. Additionally, fruit juices have less dietary fiber than whole fruits and contribute to an increased density of fructose [36]. During manufacturing juice, sugar is commonly added, and sugar intake has been known to increase oxidative stress through ROS generation [37]. Our previous studies also showed that patients with ALS who consumed less dietary fiber than the recommended intake had lower ALSFRS-R scores [6], and that dietary fiber was negatively related to ΔFS in patients with ALS through anti-inflammatory effects [5]. Spagnuolo et al. [38] reported that fructose consumption could influence brain function by promoting neuroinflammation and mitochondrial dysfunction in animal models of neurodegenerative diseases, such as Parkinson’s disease and ALS. Thus, it is necessary to develop the vegetable juices contained more fiber and antioxidant nutrients, but less sugar, for ALS patients. In addition, Pupillo et al. [39] showed that intake of citrus fruits was significantly associated with a reduced risk of ALS onset, but other fruits were not. The intake of citrus fruits, such as tangerines and oranges, was less than 7% of the total fruit intake in the present study. According to the Korea National Health and Nutrition Examination Survey, Koreans mostly consumed fruits such as apples, pears, bananas, and vegetables such as garlic, green onions, and onions [29]. However, the National Health and Nutrition Examination Survey in the USA reported that the most consumed fruits and vegetables were apples, bananas, oranges, lettuce, tomatoes, and carrots [40], suggesting that types of fruits and vegetables consumed by Koreans and Americans are different.

Fruits and vegetables contain a similar amount of vitamin C, but vitamin C is reduced by up to 54% during cooking processes [41,42]. Because the participants in the present study consumed raw fruits, but cooked vegetables, the intake of vitamin C was mostly from fruits. Previous studies have reported that vitamin C was not associated with prolonged survival in patients with ALS [11], nor was it associated with delayed progression in ALS mice [43].

Vegetable intake was positively associated with ALSFRS-R scores in patients with ALS [8]. Consistently, our previous studies showed that patients with ALS who consumed more vegetables had higher ALSFRS-R scores [6], and that dietary fiber from vegetables was negatively related to ΔFS and shorter survival time in patients with ALS [5]. Epidemiologic studies reported that vegetable DTAC was negatively associated with the risk of developing hypertension [44] and breast cancer [45]. In addition, vegetable consumption was negatively associated with the disease progression rate in patients with neurodegenerative diseases, such as Parkinson’s disease [46] and multiple sclerosis [47]. Flavonoids, particularly flavanols and flavones, are more abundant in vegetables than fruits [41] and accounted for more than 56% of vegetable DTAC in the present study. In ALS mice, supplementation with flavanols significantly extended the lifespan by protecting cells from ROS damage [48]. The intake of carotenoids, abundant in yellow-orange vegetables such as carrots, was also associated with better ALS function evaluated by the ALSFRS-R score [8]. Bond et al. [10] showed that regular use of carotenoid supplements was associated with longer survival duration in patients with ALS. Moreover, the spinal motor neurons of rats treated with β-carotene had significantly longer neurite lengths, suggesting that β-carotene protected against growth inhibitory effects in spinal motor neurons in ALS [49].

Consistent with the results of the legume DTAC in this study, Nieves et al. [8] reported that dietary isoflavones and food groups, including soy foods and legumes, were positively associated with ALSFRS-R in patients with ALS. In ALS mice, treatment with genistein, an isoflavone, prolonged the lifespan and increased the survival rate by suppressing the inflammatory response [50]. Moreover, soy isoflavone supplementation significantly inhibited oxidative stress and neuroinflammation in mice with Alzheimer’s disease, suggesting a protective role of isoflavones against neurodegeneration [51]. Supplementation with soy isoflavones also improved performance in several cognitive functions, including visual-spatial memory, construction, verbal fluency, and speeded dexterity in older adults [52]. Legumes are the major source of vitamin E, and regular use of vitamin E is negatively associated with a shorter survival time and mortality risk in patients with ALS [10,11]. A previous clinical trial also showed that vitamin E supplementation slowed the progression rate in patients with ALS using the ALS Health State scale [12].

In the present study, the participants in the upper third of DTAC had longer sun exposure. It was investigated that the vitamin D levels, reflecting sun exposure, had a significant effect on motor neuron survival and could be associated with survival time based on its involvement in potentiating neurotrophic factors and protecting motor neurons [53].

To the best of our knowledge, this is the first study to examine the association between DTAC and ΔFS and survival time in patients with ALS and to assess the overall dietary antioxidant capacity of foods rather than evaluating a single antioxidant effect on the prognosis of ALS. However, this study has several limitations. First, dietary intake was measured only once at the time of the survey, which might not reflect the long-term dietary intake of patients. Second, although we used databases from Korea and the United States for DTAC estimation, the antioxidant content of foods might be affected by cuisine, geographic location, climate, and growing conditions of the crop, which could overestimate or underestimate DTAC content. Third, this was a retrospective study, which did not include a control group. Moreover, the study design did not collect prospective data by controlling participants’ diets during the follow-up. Considering the rapid progression of ALS, with only a 3~5 year survival rate, it was difficult to utilize a conventional clinical trial design by controlling the diet over an extended period. Future prospective studies with a large ALS cohort that controls dietary patterns should be conducted. However, investigating dietary characteristics from an ALS cohort in a specialized clinic, combining the data with our previous research including DTAC, consuming foods rich in antioxidants could potentially benefit ALS patients’ dietary management. Lastly, although adjustments were made for potential confounders, unmeasured residual factors could have affected the outcomes in the present study.

## 5. Conclusions

The present study suggests that the intake of antioxidants, particularly those derived from vegetables and legumes, might have a beneficial effect on delaying disease progression and prolonging survival in Korean patients with ALS. In addition, further studies with large prospective cohorts and clinical trials are needed to confirm whether the high DTAC from vegetables and legumes improves the prognosis of ALS.

## Figures and Tables

**Figure 1 nutrients-14-03264-f001:**
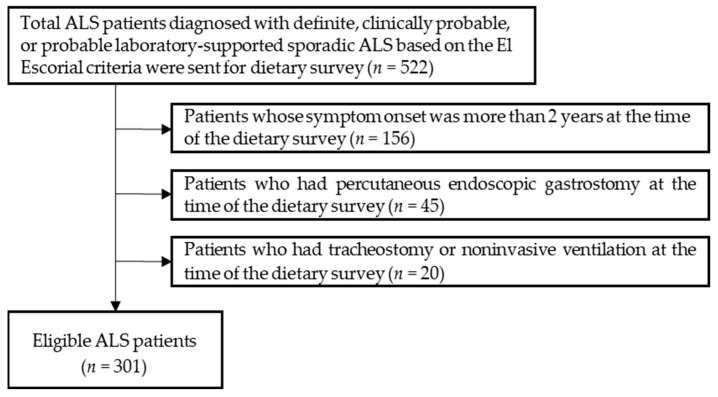
Flow chart for the selection of participants with amyotrophic lateral sclerosis (ALS).

**Figure 2 nutrients-14-03264-f002:**
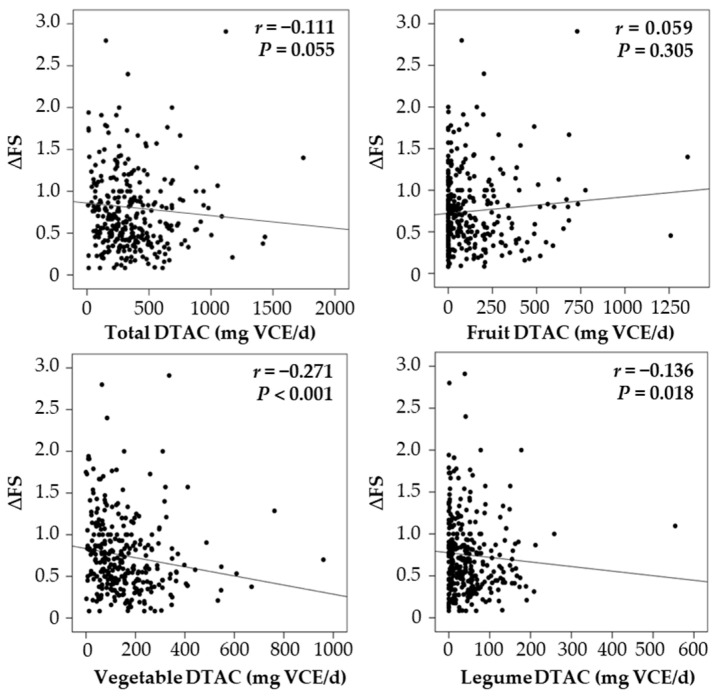
Association between dietary total antioxidant capacity (DTAC) and disease progression rate (ΔFS). The correlation between DTAC and ΔFS was determined by Spearman’s rank correlation coefficients.

**Figure 3 nutrients-14-03264-f003:**
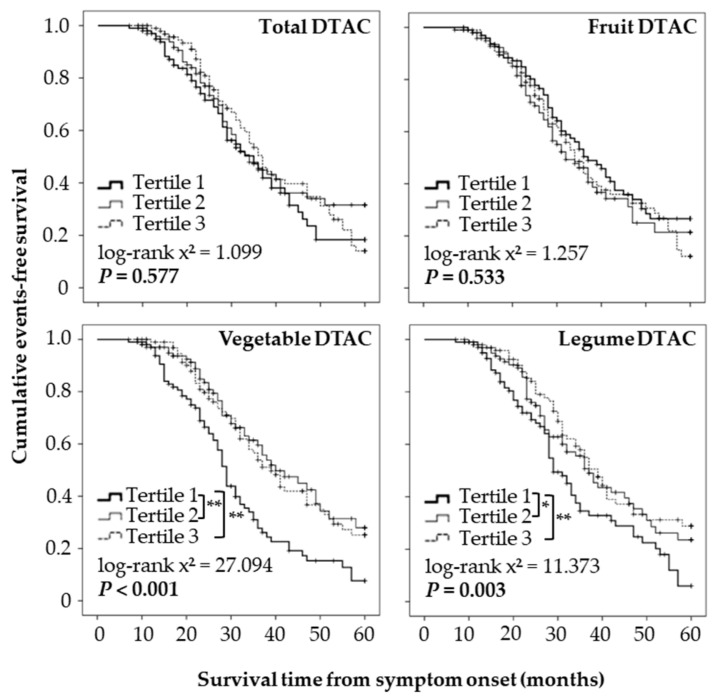
Kaplan–Maier survival curves showing 60-month survival according to the tertiles of the dietary total antioxidant capacity (DTAC). Statistical significance was determined using the log-rank test and the Kaplan–Meier method. * *p* < 0.05, ** *p* < 0.01.

**Figure 4 nutrients-14-03264-f004:**
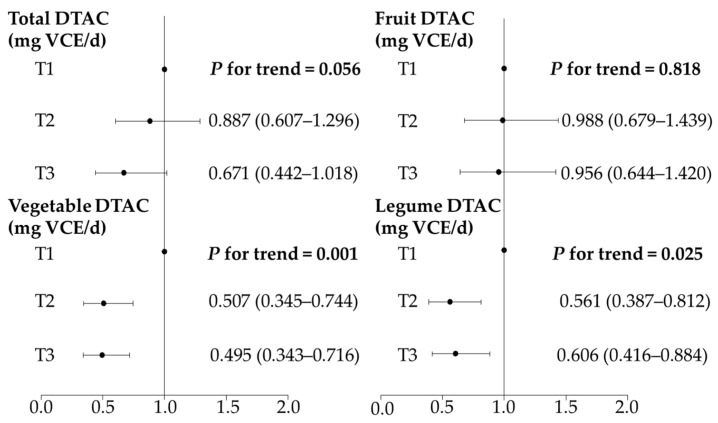
Cox proportional hazards regression analysis of event status in participants with amyotrophic lateral sclerosis (ALS) according to the tertiles of dietary total antioxidant capacity (DTAC). The adjusted hazard ratio (HR) and 95% confidence interval (CI) were determined by Cox proportional hazards regression analysis after adjusting for the age at symptom onset, sex, body mass index, onset site, disease progression rate, and energy intake.

**Table 1 nutrients-14-03264-t001:** Demographics and clinical features of the participants according to tertiles of dietary total antioxidant capacity (DTAC) ^1^.

Variables	Total (*n* = 301)	Tertiles of Total DTAC (mg VCE/d)			*p*-Value ^2^
		Total T1 (*n* = 100) ≤228.73	Total T2 (*n* = 101) 228.74–396.77	Total T3 (*n* = 100) >396.77	
DTAC (mg VCE/d)	358.23 ± 256.94	127.82 ± 62.29 ^3^^a^	307.97 ± 50.22 ^b^	639.42 ± 239.91 ^c^	<0.001
Type of endpoint, *n* (%)					0.539
PEG	90 (29.9)	33 (33.0)	31 (30.7)	26 (26.0)	
Tracheostomy	47 (15.6)	17 (17.0)	14 (13.9)	16 (16.0)	
Death	37 (12.3)	9 (9.0)	12 (11.9)	16 (16.0)	
Age at symptom onset (y)	54.64 ± 10.43	55.69 ± 11.10	54.11 ± 10.20	54.12 ± 9.98	0.384
Sex, male, *n* (%)	158 (52.5)	54 (54.0)	54 (53.5)	50 (50.0)	0.828
Bulbar onset, *n* (%)	66 (21.9)	26 (26.0)	15 (14.9)	25 (25.0)	0.107
Symptom duration (months) ^4^	15.30 ± 4.91	16.12 ± 4.63	15.16 ± 5.22	14.62 ± 4.79	0.102
ΔFS ^5^	0.75 ± 0.46	0.81 ± 0.47	0.73 ± 0.45	0.71 ± 0.46	0.203
ALSFRS-R score (0–48)	37.35 ± 6.23	35.84 ± 6.70 ^3^^a^	37.83 ± 5.70 ^ab^	38.38 ± 6.01 ^b^	0.005
Bulbar score (0–12)	9.97 ± 1.97	9.60 ± 2.08	10.18 ± 1.88	10.12 ± 1.90	0.062
BMI (kg/m^2^)	22.68 ± 2.88	22.14 ± 2.99	22.94 ± 2.61	22.96 ± 2.96	0.068
Exercise, *n* (%)	197 (65.4)	54 (54.0)	73 (72.3)	70 (70.0)	0.012
Sun exposure, *n* (%)					0.008
Never	79 (26.2)	39 (39.0)	21 (20.8)	19 (19.0)	
<30 min	87 (28.9)	25 (25.0)	34 (33.7)	28 (28.0)	
≥30 min	135 (44.9)	36 (36.0)	46 (45.5)	53 (53.0)	
Smoking, *n* (%)	109 (36.2)	33 (33.0)	41 (40.6)	35 (35.0)	0.509
Drinking, *n* (%)	144 (47.8)	52 (52.0)	50 (49.5)	42 (42.0)	0.337
Treated with riluzole, *n* (%)	270 (89.7)	89 (89.0)	90 (89.1)	91 (91.0)	0.391
Treated with edaravone, *n* (%)	32 (10.7)	10 (10.0)	14 (13.9)	8 (8.0)	0.872

VCE, vitamin C equivalent; PEG, percutaneous endoscopic gastrostomy; ΔFS, disease progression rate; ALSFRS-R, amyotrophic lateral sclerosis functional rating scale-revised; BMI, body mass index. ^1^ Values are presented as mean ± SD or number of participants (percentage distribution), as appropriate. ^2^
*p*-values were calculated using the Kruskal–Wallis test for DTAC, age at symptom onset, time point of dietary survey from symptom onset, ASLFRS-R score, and ΔFS or one-way ANOVA for BMI, followed by Bonferroni’s post hoc test for continuous variables. ^3^ Values with different superscript letters (i.e., a, b, c) in the same row are significantly different at *p* < 0.05, according to the ranked ANOVA with Bonferroni’s post hoc test. ^4^ Symptom duration (months) refers to the time point of the dietary survey from symptom onset. ^5^ ΔFS was calculated using the following formula: ΔFS = (48 − ALSFRS-R score at the time of survey/duration from symptom onset to the time of the survey (months)).

**Table 2 nutrients-14-03264-t002:** Demographics and clinical features of the participants according to tertiles of the dietary total antioxidant capacity (DTAC) of major food groups ^1^.

Variables	T1 (*n* = 100)	T2 (*n* = 101)	T3 (*n* = 100)	*p*-Value ^2^
	Fruit DTAC (mg VCE/Day)			
	≤12.62	12.63–126.55	>126.55	
Age at symptom onset (year)	53.86 ± 11.76	55.67 ± 10.32	54.37 ± 9.05	0.447
Sex, male, *n* (%)	61 (61.0)	53 (52.5)	44 (44.0)	0.055
Bulbar onset, *n* (%)	20 (20.0)	21 (20.8)	25 (25.0)	0.562
Symptom duration (months) ^4^	15.93 ± 4.88	15.46 ± 5.06	14.51 ± 4.72	0.101
ΔFS ^5^	0.68 ± 0.42 ^3a^	0.82 ± 0.45 ^b^	0.75 ± 0.51 ^b^	0.033
ALSFRS-R score (0–48)	37.85 ± 6.41	36.24 ± 6.10	37.98 ± 6.07	0.086
Bulbar score (0–12)	10.13 ± 2.03	9.80 ± 2.09	9.97 ± 1.77	0.499
	Vegetable DTAC (mg VCE/day)			
	≤83.92	83.93–170.06	>170.06	
Age at symptom onset (year)	54.17 ± 10.18	56.27 ± 9.88	53.46 ± 11.09	0.139
Sex, male, *n* (%)	56 (56.0)	49 (48.5)	53 (53.0)	0.564
Bulbar onset, *n* (%)	21 (21.0)	24 (23.8)	21 (21.0)	0.959
Symptom duration (months) ^4^	14.62 ± 4.46	15.99 ± 5.03	15.28 ± 5.15	0.132
ΔFS ^5^	0.93 ± 0.49 ^3a^	0.70 ± 0.41 ^b^	0.62 ± 0.42 ^b^	<0.001
ALSFRS-R score (0–48)	35.24 ± 6.78 ^3a^	37.66 ± 5.59 ^b^	39.15 ± 5.67 ^b^	<0.001
Bulbar score (0–12)	9.69 ± 2.01	9.88 ± 1.83	10.33 ± 2.02	0.061
	Legume DTAC (mg VCE/day)			
	≤15.26	15.27–53.89	>53.89	
Age at symptom onset (year)	55.97 ± 9.36	54.61 ± 11.64	53.33 ± 10.07	0.202
Sex, male, *n* (%)	54 (54.0)	48 (47.5)	56 (56.0)	0.453
Bulbar onset, *n* (%)	22 (22.0)	21 (20.8)	23 (23.0)	0.911
Symptom duration (months) ^4^	14.99 ± 4.42	15.62 ± 5.06	15.28 ± 5.23	0.658
ΔFS ^5^	0.83 ± 0.50 ^3a^	0.76 ± 0.47 ^ab^	0.65 ± 0.39 ^b^	0.020
ALSFRS-R score (0–48)	36.09 ± 7.07 ^3a^	37.20 ± 6.09 ^ab^	38.77 ± 5.13 ^b^	0.016
Bulbar score (0–12)	9.70 ± 2.18	9.91 ± 1.74	10.29 ± 1.93	0.058

VCE, vitamin C equivalent; ΔFS, disease progression rate; ALSFRS-R, amyotrophic lateral sclerosis functional rating scale-revised. ^1^ Values are presented as mean ± SD or number of participants (percentage distribution), as appropriate. ^2^
*p*-values were calculated using the Kruskal–Wallis test for DTAC, time point of dietary survey from symptom onset, ASLFRS-R score, and ΔFS or one-way ANOVA for age at symptom onset, followed by Bonferroni’s post hoc test for continuous variables. ^3^ Values with different superscript letters (i.e., a, b) in the same row are significantly different at *p* < 0.05, according to the ranked ANOVA with Bonferroni’s post hoc test. ^4^ Symptom duration (months) refers to the time point of the dietary survey from symptom onset. ^5^ ΔFS was calculated using the following formula: ΔFS = (48 − ALSFRS-R score at the time of survey/duration from symptom onset to the time of the survey (months)).

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
