# Peer review of "Relationship between Dietary Total Antioxidant Capacity and the Prognosis of Amyotrophic Lateral Sclerosis"

_nutrients, 2022, doi:10.3390/nu14163264_

Round 1
Reviewer 1 Report
The manuscript “Relationship between dietary total antioxidant capacity and the prognosis of amyotrophic lateral sclerosis” described antioxidants from vegetables and legumes have beneficial effects in slowing disease progression and prolonging survival in patients with ALS. This article is generally a good clinical guide. However, there are some issues that need to be revised. Detailed questions are given below.
1. ΔFS seems to be non-significantly correlated with total DTAC, so how can the data be characterized by this indicator? Please highlight and explain?
2. It is necessary and interesting to discuss juices and fresh fruit because of swallowing disorders. But the explanation from the point of view of dietary fiber seems a bit forced, or one of the reasons. Is it possible to suggest a further discussion in terms of the vitamin content of fruit juices and fresh fruit? Would this be a more reasonable formulation?
3. The language expression in the essay needs to be improved, please get a native English speaker to revise it, and some sentences are too long.
4. Please recheck the format of the references. There are many missing full-page numbers.
Author Response
Point 1: ΔFS seems to be non-significantly correlated with total DTAC, so how can the data be characterized by this indicator? Please highlight and explain?
Response: Thank you for your comment. In this study, we emphasized that even though ΔFS did not correlate with Total DTAC, it correlated with DTAC from vegetables and legumes. This data suggests that the composition of nutrients in each food group, such as vegetables and legumes, important for disease progression speed(ΔFS).
DTAC was calculated according to the total antioxidant capacity (mg VCE) multi-plied by the intake of each food (g) estimated from 24-h recall. Then, DTAC was categorized into the 17 food groups suggested by the Korea National Health and Nutrition Examination Survey. A previous study also showed inconsistent results when comparing total food intake with the intake from each food group (Yu et al. 2020, Relationship between dietary fiber intake and the prognosis of amyotrophic lateral sclerosis in Korea.) We think this would be related to the difference in nutritional compositions in each food groups. Therefore, we analyzed and compared three subgroup data. Out main finding was that the relation between the DTAC from vegetables and legumes and the disease progression. Additionally, our results suggest that nutrients in vegetables and legumes which are not counted in DTAC calculation might have beneficial effects on ΔFS.
Point 2: It is necessary and interesting to discuss juices and fresh fruit because of swallowing disorders. But the explanation from the point of view of dietary fiber seems a bit forced, or one of the reasons. Is it possible to suggest a further discussion in terms of the vitamin content of fruit juices and fresh fruit? Would this be a more reasonable formulation?
Response: Thank you for your suggestion. According to previous studies, fruit juices have lower antioxidants, higher sugar content, and lower dietary fiber than fresh fruits. We stated in the discussion section that consumption of these commercial fruit juices and fruit in the form of fruit juice would not be beneficial for the patient's prognosis, even though juice is a favored form and easily consumed than solid fruit by patients with dysphagia. Thus, we added a suggestion that it is necessary to develop vegetable juices containing more fiber and antioxidants but less sugar, so they can be safely consumed by patients with dysphasia, and also rich in good nutrients.
This point was reflected in the discussion section as follows:
- Van der Sluis et al. [35] reported that apple juice contained less antioxidants, particu-larly flavonoids than fresh apples, which were not transferred into the juice and re-mained in the pulp. Additionally, fruit juices have less dietary fiber than whole fruits and contribute to an increased density of fructose [36]. During manufacturing juice, sugar is commonly added, and sugar intake has been known to increase oxidative stress through ROS generation [37]. Our previous studies also showed that patients with ALS who consumed less dietary fiber than the recommended intake had lower ALSFRS-R scores [6], and that dietary fiber was negatively related to ∆FS in patients with ALS through anti-inflammatory effects [5]. Spagnuolo et al. [38] reported that fructose consumption could influence brain function by promoting neuroinflammation and mitochondrial dysfunction in animal models of neurodegenerative diseases, such as Parkinson’s disease and ALS. Thus, it is necessary to develop the vegetable juices contained more fiber and antioxidant nutrients but less sugar for ALS patients.
Point 3: The language expression in the essay needs to be improved, please get a native English speaker to revise it, and some sentences are too long.
Response: Thank you for your comment. We have the manuscript revised professionally by a native English speaker.
Point 4: Please recheck the format of the references. There are many missing full-page numbers.
Response: Thank you for pointing out. We have rechecked and revised the references.
Reviewer 2 Report
The authors have tried to correlate the total antioxidant in the level on the prognosis of ALS which is an interesting study, but the study to definitely prove the hypothesis.
1. The study does not have any controls. If the authors have gathered data from the clinic, it would have been easy for them to gather some data for patients that would serve as controls.
2. The effect of dietary antioxidants is well known and the study is based on that, but here the patients who were recruited were not given the same diet to be followed. Without this it is not possible to come to a conclusion based on a questionnaire that’s was obtained from the clinic.
3. The important point which the authors missed is the effect of the drugs that the patients were prescribed for their condition
4. The quantity of food that each patient took in each type to account for the antioxidant levels in the food is not given. May be a patient consumed some food in large quantities than compared with the others. This is not a reflection of the antioxidant levels of the patients.
5. Unless all the patients followed the same dietary pattern a conclusion cannot be made as the authors have stated in the paper
Round 2
Reviewer 2 Report
The authors have not fully addressed the answers, but have done their best to incorporate the necessary details that carries some merit to the paper. I hope the authors would provide appropriate controls and the monitor the diet regime, if they are going to undertake any studies like this in the future.